# Mechanical properties of raw bamboo fiber-reinforced phosphogypsum-based composite cementitious materials and their strengthening mechanism

Xian Fu⊕, Peng Liu⊕⊕*, Dewen Kong⊕, Yuan Li⊕, Yongfa Wang⊕

College of Civil Engineering, Guizhou University, Guiyang, Guizhou, China

⊕ These authors also contributed equally to this work.
* liupeng00865032@163.com

## Abstract

Phosphogypsum-based cementitious materials (PGCs) typically exhibit low strength and poor toughness when utilized as construction materials. This study explores the incorporation of raw bamboo fibers of varying lengths into PGCs at different ratios to develop raw bamboo fiber-reinforced phosphogypsum-based composite materials (BFRPGCs). Firstly, the influence of the water-cement ratio on the mechanical properties of BFRPGCs was investigated through a one-factor experimental approach, leading to the identification of an optimal water-cement ratio. Secondly, the effects of the length and content of raw bamboo fibers on the mechanical properties of BFRP-GCs at this optimal water-cement ratio were examined, along with an exploration of the mechanisms by which raw bamboo fibers impact the mechanical properties of the composites, considering their damage modes. Finally, the microstructure of BFRP-GCs was analyzed using scanning electron microscopy (SEM), which elucidated the mechanisms through which raw bamboo fibers influence the working and mechanical properties of the composites. The results demonstrated that the incorporation of raw bamboo fibers significantly enhanced the mechanical strength of the specimens. Specifically, when the length of the bamboo fibers was 12 mm and the doping amount was 1.0%, the compressive and flexural strengths of BFRPGCs reached their maxima at 28.99 MPa and 8.41 MPa, respectively. These values represent increases of 123.73% and 169.82% compared to the control group. Additionally, hydration-generated calcium silicate hydrate (C-S-H) gels and calcium aluminate (AFt) phases formed a skeletal support around the $CaSO_4 \cdot 2H_2O$, reinforcing the matrix structure. Furthermore, numerous hydration products adhered to the surfaces of the raw bamboo fibers, resulting in enhanced adhesion between the fibers and the matrix. This study provides valuable insights for the research and application of fiber-reinforced phosphogypsum-based building materials.

**Data availability statement:** All experimental data files are available from the Figshare database (accession number 10.6084/m9.figshare.28130249).

**Funding:** Guizhou Provincial Basic Research Program (Natural Science) (Qiankehejichu-ZK[2023] General 067 to PL); the Guizhou Provincial Basic Research Program (Natural Science) (Qiankehejichu MS[2025] 677 to PL); the Innovation Fund of Guizhou University Survey and Design Institute Co. Design Institute Co., Ltd. (Guidakancha [2022] 05 to YL).

**Competing interests:** The authors have declared that no competing interests exist.

## 1. Introduction

Phosphogypsum is a byproduct generated during phosphate production, primarily composed of calcium sulfate dihydrate. The substantial accumulation of phosphogypsum, coupled with severe environmental pollution and low utilization rates, underscores the urgent need for effective recycling and reuse strategies [1–3]. As of 2024, China's stockpile of phosphogypsum has surpassed 600 million tons, increasing by over 80 million tons annually, with storage facilities approaching capacity [4–6]. The comprehensive utilization of phosphogypsum encompasses three main avenues: 1) its application as a construction material [7–9], 2) its use in agriculture [10–12], and 3) its decomposition for sulfuric acid production, which is essential for cement co-production [13–15]. Among these, the use of phosphogypsum in construction has gained prominence, yet the inherent limitations of phosphogypsum-based building materials, characterized by poor toughness and low strength, hinder its widespread adoption in the construction sector [4,16,17].

To enhance the mechanical properties of phosphogypsum and broaden its construction applications, numerous studies have been conducted globally. However, practical engineering applications face challenges: steel fibers are prone to corrosion when exposed to air [18,19], polypropylene (PP) fibers exhibit low tensile strength [20,21], polyvinyl alcohol fibers are costly [22,23], and glass and asbestos fibers pose health hazards [24,25]. In contrast, plant fibers are natural polymer materials with specific chemical compositions [26,27], drawing significant attention due to their safety, availability, biodegradability, and potential for sustainable regeneration [28–30]. Among these, bamboo fiber—a rapidly renewable plant fiber—exhibits mechanical properties comparable to traditional fiber-reinforced cement, significantly enhancing the toughness of composite cementitious materials [31,32]. Moreover, bamboo fibre-reinforced phosphogypsum has been demonstrated to exhibit distinct advantages over conventional fibre-reinforced materials with regard to environmental friendliness, resource and sustainability, and compatibility with biodegradable materials.

Current research on bamboo fibers focuses on their application as a substitute for traditional fiber-reinforced geopolymers and cementitious materials. For instance, Correia et al. [33] demonstrated that bamboo fibers could effectively reinforce cement, resulting in mechanical properties similar to those of conventional fiber-reinforced composites. Zhou et al. [34] introduced a novel bamboo fiber-reinforced phosphogypsum composite wall, highlighting the synergistic benefits of raw bamboo and phosphogypsum for green building applications. Bala et al. [35] explored bamboo reinforcement in concrete, finding that an admixture of approximately 4% bamboo significantly improved the material's strength and toughness while minimizing cracking. Liu et al. [36] developed a stress-strain principal model for raw bamboo and phosphogypsum, proposing a calculation method for the load-bearing capacity of combined short columns, supported by experiments involving bamboo cylinders filled with phosphogypsum and finite element analysis. While bamboo is widely utilized in the construction industry, its composite products with gypsum are largely limited to gypsum walls, panels, and bricks. Consequently, there is a notable lack of research on raw bamboo

fiber-reinforced phosphogypsum-based composite cementitious materials. Exploring the potential of raw bamboo fibers to reinforce phosphogypsum cementitious materials could significantly improve the overall utilization rate of phosphogypsum.

In this study, we prepared raw bamboo fiber-reinforced phosphogypsum-based composite materials (BFRPGCs) by blending raw bamboo fibers with phosphogypsum-based cementitious materials to optimize the properties of these composites. A one-way experimental design was employed to investigate the effect of the water-cement ratio on BFRPGCs, identifying the optimal ratio for enhanced performance. Under this optimal water-cement ratio, we analyzed the effects of fiber length, doping amount, and their interactions on composite properties. Additionally, the microscopic characteristics of bamboo fibers within phosphogypsum-based building materials were examined using scanning electron microscopy (SEM), and the mechanisms by which bamboo fibers influence the properties of phosphogypsum-based composites were explored. This research aims to provide valuable insights for the development and application of fiber-reinforced phosphogypsum-based building materials.

## 2. Experimental program

### 2.1. Material

**2.1.1. Phosphogypsum-based cementitious materials.** The principal raw materials employed in this experiment included phosphogypsum, cement, silica fume, quicklime, and a water-reducing agent. Phosphogypsum was sourced from Guizhou Kaifu Phosphogypsum Comprehensive Utilization Co., while the cement utilized was ordinary silicate cement (P.O. 42.5) produced by Guiyang Conch Panjiang Cement Plant. Silica fume, primarily composed of $SiO_2$, was obtained from Henan Gongyi Baichuan Environmental Protection Engineering Co., Ltd. Quicklime, containing over 97% active CaO, was procured from Sichuan Yibin Sichuan Fume Biotechnology Co., Ltd. Additionally, the polycarboxylic acid water-reducing agent was purchased from Shanghai Chenqi Chemical Technology Co. Both primary phosphogypsum (DPG) and hemihydrate phosphogypsum (HPG) were utilized in this study. The preparation method was as follows: the phosphogypsum was crushed, naturally dried, and sieved to obtain DPG. This DPG was then calcined in an oven at 160 °C for 2 hours, followed by a sealing process for natural aging over 7 days to yield HPG [37]. Fig 1 illustrates the X-ray diffractograms of the two types of phosphogypsum, while Table 1 presents the chemical composition of the principal raw materials.

**2.1.2. Raw bamboo fibres.** The bamboo fibers used in this study were bundled natural yellow bamboo fibers (Fig 2(a)) sourced from Sichuan Changsheng New Material Technology Co. Their performance parameters are detailed in

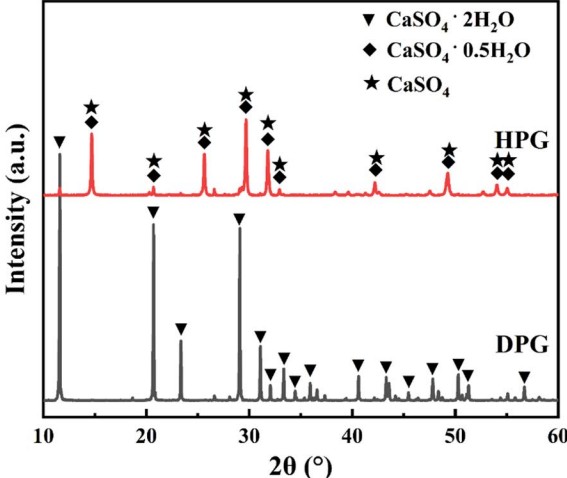

**Fig 1. XRD pattern of DPG, HPG.**

**Table 1. Detailed chemical composition of raw materials.**

| Materials | SO$_3$ | CaO | SiO$_2$ | P$_2$O$_5$ | Fe$_2$O$_3$ | Al$_2$O$_3$ | MgO | K$_2$O |
|---|---|---|---|---|---|---|---|---|
| DPG | 52.009 | 43.869 | 1.863 | 0.878 | 0.564 | 0.315 | 0.036 | 0.078 |
| HPG | 54.163 | 42.033 | 1.709 | 0.918 | 0.471 | 0.294 | 0.017 | 0.074 |
| Cement | 3.962 | 61.713 | 19.897 | 0.169 | 4.456 | 5.155 | 1.725 | 1.196 |
| Silica fume | 0.316 | 0.130 | 96.030 | 0.011 | 0.078 | 0.280 | 0.107 | 0.191 |
| Quicklime | 0.263 | 97.231 | 0.448 | 0.003 | 0.132 | 0.139 | 1.902 | 0.030 |

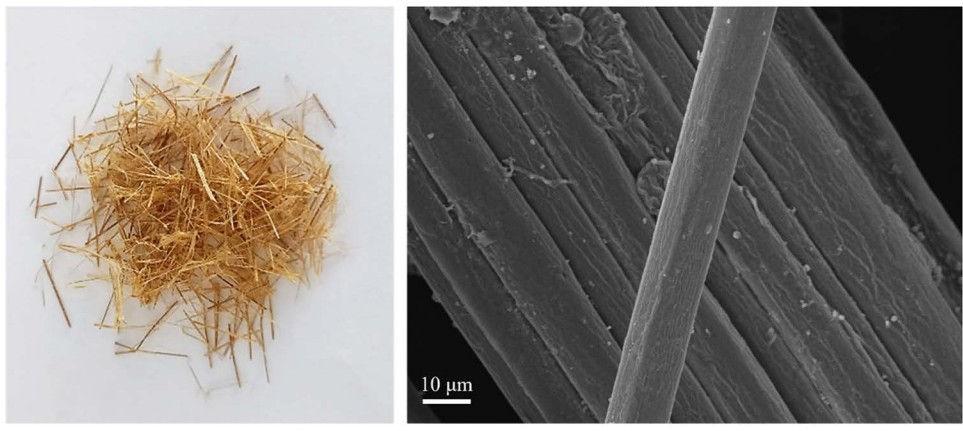

(a) Macroscopic morphology          (b) Microscopic morphology

**Fig 2. Macroscopic and microscopic morphology of raw bamboo fibres.**

Table 2, while their microscopic morphology is depicted in Fig 2(b). The lengths of the raw bamboo fibers employed in the experiments were 4 mm, 8 mm, 12 mm, and 16 mm, which were prepared through both manual and mechanical cutting methods.

## 2.2. Method

**2.2.1. Experimental design.** In the pre-experiment, we observed that when the water-cement ratio was below 0.25, the workability of PGCs was poor and difficult to handle. On the other hand, when the water-cement ratio exceeded 0.45, the strength of PGCs decreased. Based on these findings, we conducted a one-factor experiment with water-cement ratio as the influencing factor, ranging from 0.25 to 0.45. The optimal water-cement ratio of 0.275 was determined, with raw bamboo fiber length fixed at 8 mm and doping amount at 1.0%. The initial experiment employed a one-way design, focusing primarily on the water-cement ratio. The optimal water-cement ratio of 0.275 was determined while maintaining the length of the raw bamboo fibers at 8 mm and the dosage at 1.0 wt %. Following this, the effects of fiber length and dosage on the mechanical properties of bamboo fiber-reinforced polymer concrete (BFRPGCs) were investigated. The lengths of the raw bamboo fibers were 4 mm, 8 mm, 12 mm, and 16 mm, with the fiber dosage for each length set at four levels: 0, 0.50, 1.0, and 1.5 wt%. The experimental design is summarized in Table 3. In this study, samples were labeled according to the length and dosage of the raw bamboo fibers. For instance, the designation "BF8-1.0" indicates a fiber length of 8 mm with a dosage of 1.0 wt %.

**Table 2. Physical and mechanical properties of raw bamboo fibres.**

| Density (g·cm⁻³) | Average diameter (mm) | Tensile strength (MPa) | Elastic Modulus (GPa) | Elongation (%) |
|---|---|---|---|---|
| 1.3-1.5 | 0.18 | 350-800 | 25-45 | 2.5-5.8 |

Density (g·cm$^{-3}$)

**Table 3. Mixing ratio design for BFRPGCs.**

| Mix Designation | Powder proportion (wt.%) | | Cement | Silica fume | Quicklime | Fiber Length/mm | Fiber Length/% |
|---|---|---|---|---|---|---|---|
| | PG | | | | | | |
| | DPG | HPG | | | | | |
| BF00 | 60 | 40 | 15 | 5 | 4 | 0 | 0 |
| BF4-0.5 | 60 | 40 | 15 | 5 | 4 | 4 | 0.5 |
| BF4-1.0 | 60 | 40 | 15 | 5 | 4 | 4 | 1.0 |
| BF4-1.5 | 60 | 60 | 15 | 5 | 4 | 4 | 1.5 |
| BF8-0.5 | 60 | 40 | 15 | 5 | 4 | 8 | 0.5 |
| BF8-1.0 | 60 | 40 | 15 | 5 | 4 | 8 | 1.0 |
| BF8-1.5 | 60 | 40 | 15 | 5 | 4 | 8 | 1.5 |
| BF12-0.5 | 60 | 60 | 15 | 5 | 4 | 12 | 0.5 |
| BF12-1.0 | 60 | 60 | 15 | 5 | 4 | 12 | 1.0 |
| BF12-1.5 | 60 | 40 | 15 | 5 | 4 | 12 | 1.5 |
| BF16-0.5 | 60 | 40 | 15 | 5 | 4 | 16 | 0.5 |
| BF16-1.0 | 60 | 40 | 15 | 5 | 4 | 16 | 1.0 |
| BF16-1.5 | 60 | 60 | 15 | 5 | 4 | 16 | 1.5 |

**2.2.2. Specimen making.** The mass ratio of DPG to HPG was set at 60:40, with the remaining components comprising 15% cement, 5% silica fume, and 4% quicklime, relative to the total mass of the polymer grout (PG). Additionally, the bamboo fibers used in each experimental trial were weighed according to the specifications outlined in Table 3. A water-reducing agent was consistently mixed at a concentration of 0.2% across all experiments, and the water-to-material ratio was established at 0.275. Initially, the aforementioned powders were combined and blended thoroughly. Water was then added, followed by blending and stirring: first at a slow speed for 30 seconds, and subsequently at a high speed for 90 seconds using an electric mixer, to achieve a homogeneously mixed slurry. This slurry was poured into a mold with dimensions of 40 mm×40 mm×160 mm and was removed from the mold once the final setting had occurred. The specimens underwent curing for 7 days and 28 days under natural conditions at a temperature of 20±2 °C. Following the curing periods, a series of performance tests were conducted on the specimens. All physical and mechanical property experiments were performed at the School of Civil Engineering Testing Center at Guizhou University.

## 2.3. Experimental test methods

Upon completion of the specified maintenance procedures, the samples will be transferred to an oven set at a temperature of 40±2 °C until their quality stabilizes. Subsequently, the specimens will undergo mechanical properties testing [38]. The determination of compressive and flexural strength of bamboo fiber-reinforced polymer concrete (BFRPGCs) will be conducted in accordance with GB/T 17669.3-1999, titled "Determination of Mechanical Properties of Building Gypsum" [39].

**2.3.1. Compressive strength.** The 7d and 28d compressive strength tests were conducted using a YA-300 microcomputer-controlled electro-hydraulic servo pressure tester, produced by Changchun Kexin Testing Instruments Co.

 

**2.3.2. Flexural strength.** The flexural strength tests, conducted on a 7-day and 28-day timescale, were performed using a DKZ-5000 electric flexural tester, produced by Jianyi Zhongke Instruments.

**2.3.3. Micromorphological analysis.** The micro-morphological observations of the specimens were conducted using a TESCAN MIRA LMS scanning electron microscope, manufactured in the Czech Republic.

## 3. Results and discussion

### 3.1. Effect of water-cement ratio on mechanical properties of BFRPGCs

Fig 3 illustrates the effect of the water-cement ratio on the compressive strength of bamboo fiber-reinforced polymer concrete (BFRPGCs). As shown in the figure, the 7-day and 28-day compressive strengths of BFRPGCs gradually decrease as the water-cement ratio increases from 0.250 to 0.450. At a water-cement ratio of 0.250, the 7-day compressive strength is 19.02 MPa, and the 28-day compressive strength is 27.08 MPa, representing the maximum values. In contrast, at a water-cement ratio of 0.450, the 7-day compressive strength drops to 1.66 MPa, and the 28-day compressive strength decreases to 2.41 MPa, indicating the minimum values. Increasing the water-cement ratio provides an adequate water environment for the hydration process of phosphogypsum-based cementitious materials, allowing the hydration reaction to proceed smoothly. However, excessive water can lead to issues such as water secretion and delamination of the material during the hardening process [40]. These phenomena compromise the internal structure of the matrix, leading to the formation of defects such as cracks and voids. Additionally, the evaporation of excess water creates more pores, resulting in increased porosity as the water-cement ratio continues to rise. This leads to a less dense internal cementation of the specimen and causes stress concentration when subjected to pressure, which is macroscopically manifested as a reduction in compressive strength.

Fig 4 demonstrates the influence of the water-cement ratio on the flexural strength of BFRPGCs. The data indicate that the flexural strengths at both 7 days and 28 days decrease progressively as the water-cement ratio increases from 0.250 to 0.450. Specifically, at a water-cement ratio of 0.250, the 7-day flexural strength reaches 6.71 MPa, while the 28-day flexural strength peaks at 8.05 MPa, representing the highest values recorded. In contrast, at a water-cement ratio of 0.450, the 7-day flexural strength falls to 1.54 MPa, and the 28-day strength declines to 1.85 MPa, marking the lowest values observed. As the water-cement ratio increases, the higher water content leads to a greater retention of free water during the hardening process of phosphorite-based cementitious materials. The evaporation of this free water results in

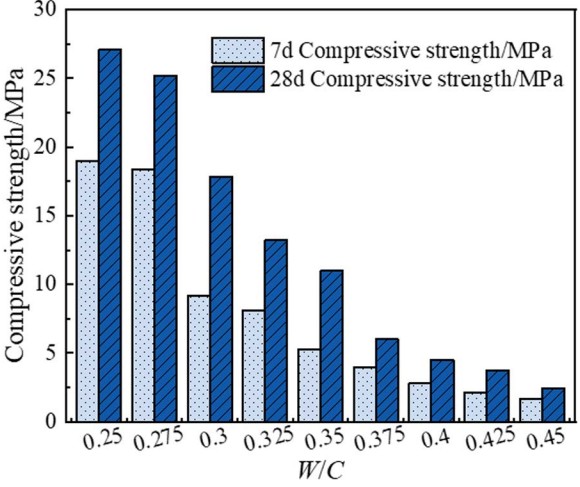

**Fig 3. Effect of water-cement ratio on compressive strength of BFRPGCs.**

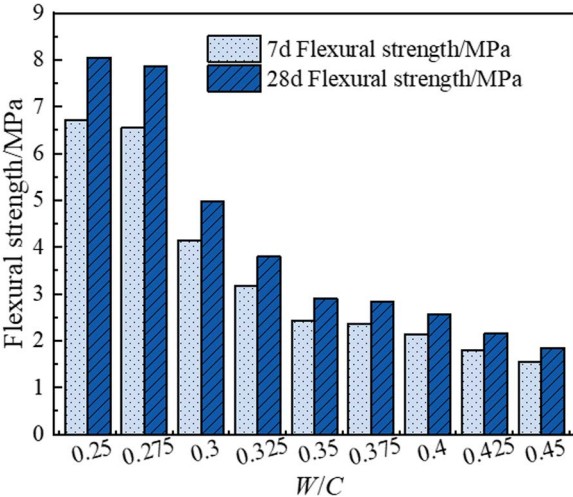

**Fig 4. Effect of water-cement ratio on flexural strength of BFRPGCs.**

pore formation, which subsequently increases the material's porosity. An increase in porosity compromises the internal structure of the phosphogypsum matrix, leading to a reduction in density. Consequently, the specimens become more prone to fracture under external loads. Additionally, the increase in porosity, combined with a more heterogeneous distribution of voids, weakens the bonding strength within the phosphogypsum matrix. This further enhances the material's vulnerability to fracture when subjected to bending moments.

In addition, the experiments revealed a delamination phenomenon in the bamboo fiber-reinforced phosphogypsum composite cementitious material when the water-cement ratio exceeded 0.325. At a water-cement ratio of 0.250, the formation of this composite material was hindered, leading to a tendency for bamboo fibers to agglomerate. Consequently, it is essential to maintain a moderately high water-cement ratio to prevent low strength and delamination while ensuring that it is not too low to avoid mixing difficulties. Although the compressive strength at 7 days and 28 days for a water-cement ratio of 0.275 was found to be 3.36% and 6.87% lower, respectively, than that at a water-cement ratio of 0.250, the flexural strength at 7 days and 28 days was observed to be 2.39% and 2.36% lower than at the lower ratio. A comparison of Fig 3 and Fig 4 indicates that the reductions in compressive and flexural strengths at a water-cement ratio of 0.275 are minor compared to those observed at 0.250. However, once the water-cement ratio exceeds 0.300, a significant decline in both compressive and flexural strengths is noted. Thus, the optimal water-cement ratio for the bamboo fiber-reinforced phosphogypsum composite cementitious material is determined to be 0.275.

### 3.2. Effect of bamboo fibres on the mechanical properties of BFRPGCs

**3.2.1. Effect of bamboo fiber on flexural strength of BFRPGCs.** Fig 5 illustrates the variation in flexural strength of bamboo fiber-reinforced phosphogypsum composites (BFRPGCs) as a function of bamboo fiber length and doping amount. Without the addition of bamboo fibers, the 7-day and 28-day flexural strengths of PGCs were 4.31 MPa and 4.95 MPa, respectively. After the incorporation of bamboo fibers, the flexural strength of BFRPGCs gradually increased with the increase in fiber content, while the length of the bamboo fibers remained constant. In terms of 7-day flexural strength, it was observed that for a bamboo fiber length of 16 mm, the flexural strength of the specimens declined as the dosage increased. For the 28-day flexural strength, when the fiber length exceeded 8 mm, the specimens exhibited an initial increase in flexural strength followed by a decrease with rising dosage. Upon determining the optimal dosage of bamboo fibers, it was found that both the 7-day and 28-day flexural strengths of BFRPGCs initially increased

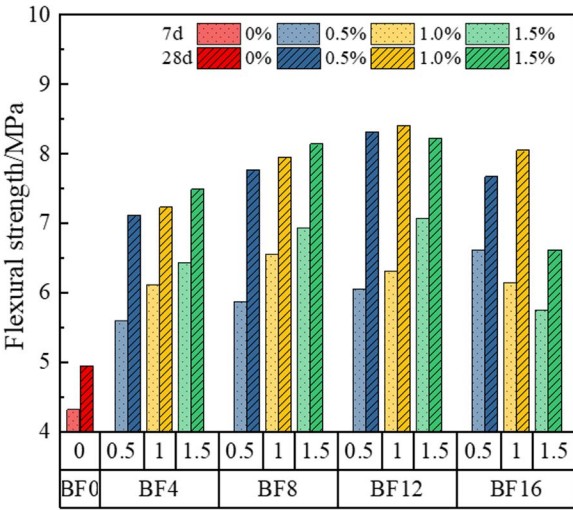

**Fig 5. Variation of flexural strength of BFRPGCs.**

before declining with longer bamboo fiber lengths. Notably, for a low dosage of 0.5%, the 7-day flexural strength of the specimens increased with longer fibers, although this trend was not particularly pronounced. For specimens with a bamboo fiber length of 12 mm, the 7-day flexural strength peaked at 7.07 MPa with a doping level of 1.5%, representing a 64.04% increase compared to the control group. For the 28-day flexural strength in specimens with a bamboo fiber length of 12 mm, the maximum value reached 8.41 MPa at a dosage of 1.0%, corresponding to a 69.90% increase relative to the control group. When the length of bamboo fibers exceeds 12 mm, the flexural strength of the samples decreases as the content increases. For the specimen with a bamboo fiber content of 1.5%, the flexural strength at 7 days and 28 days were 5.75 MPa and 6.62 MPa, respectively, which represent reductions of 18.67% and 21.28% compared to the maximum group. In conclusion, the incorporation of bamboo fibers has been shown to enhance the flexural strength of the specimens, with the optimal performance observed when the length of the bamboo fibers is 12 mm.

As illustrated in Fig 5, increasing both the length and the doping of bamboo fibers enhances the flexural strength of bamboo fiber reinforced glass composites (BFRGCs). This improvement is attributed to the longer fibers, which, when the specimen is subjected to a bending load, reinforce the bonding capacity within the matrix [41]. Additionally, higher fiber doping increases the number of interfaces, further contributing to this effect. However, it was observed that when the length of the bamboo fibers exceeded 12 mm, there was no significant enhancement in the flexural strength of the specimens at 7 and 28 days, despite the effects of fiber doping and length. This lack of improvement may result from the uneven distribution of fibers within the matrix, which can lead to defects caused by the increased fiber length and doping [42]. This observation aligns with previous studies [43,44], that concluded that the enhancement of the flexural properties of gypsum materials by fibers is primarily due to mechanical occlusion, interfacial adhesion, and cohesion between the fibers and the matrix. Moreover, a densely formed fiber-matrix interfacial transition zone (ITZ) is established between the fibers and the matrix. The superior tensile strength and fracture toughness of bamboo fibers enable a robust interface that effectively resists internal tensile stresses and absorbs substantial energy, thereby minimizing internal damage and cracking. Additionally, cracks that span the BFRGCs can serve as load bridges, enhancing the stress field within the matrix and significantly improving the overall toughness of the material [45].

**3.2.2. Effect of bamboo fiber on compressive strength of BFRPGCs.** Fig 6 illustrates the relationship between the compressive strength of Bamboo Fiber Reinforced Geopolymer Composites (BFRPGCs), the length of bamboo fibers, and the amount of doping. Without the addition of bamboo fibers, the 7-day and 28-day compressive strengths of PGCs

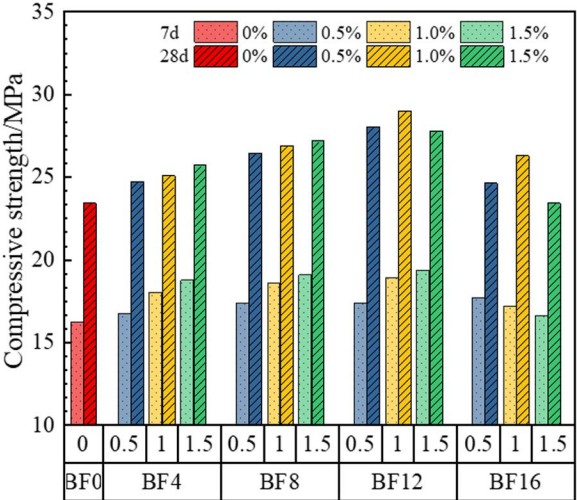

**Fig 6. Variation of compressive strength of BFRPGCs.**

were 16.20 MPa and 23.43 MPa, respectively. After the addition of bamboo fibers, the 7-day compressive strength of BFRPGCs increased gradually with the increase in fiber content, provided that the length of the bamboo fibers remained unchanged and did not exceed 12 mm. However, when the fiber length is increased to 16 mm, a decline in compressive strength occurs with the addition of doping. For the 28-day compressive strength of BFRPGCs, a similar trend is observed. When the fiber length is less than 12 mm, compressive strength increases steadily with higher doping levels. Beyond this length, the compressive strength of the specimens initially increases and then decreases with increasing doping amounts. Once the bamboo fiber dosage is set, both the 7-day and 28-day compressive strengths of BFRPGCs demonstrate a pattern of initial increase followed by a decrease as the fiber length increases. Notably, for a lower dosage of 0.5%, the compressive strengths at both 7 days and 28 days show a gradual increase with longer fibers, although this trend is less pronounced. As depicted in Fig 6, with a fixed bamboo fiber length of 12 mm and a doping level of 1.5%, the 7-day compressive strength reached a maximum of 19.35 MPa, representing a 19.44% increase compared to the control group. In specimens with a bamboo fiber length of 12 mm, the highest compressive strength for those with 1.0% doping was recorded at 28 days, achieving 28.99 MPa, which is a 23.73% increase over the control group. When the length of bamboo fibers exceeds 12 mm, the flexural strength of the samples first increases and then decreases with the increase in content. For the specimen with a bamboo fiber content of 1.5%, the compressive strength at 7 days and 28 days were 16.62 MPa and 23.42 MPa, respectively, which represent reductions of 14.11% and 19.21% compared to the maximum group. In conclusion, the addition of bamboo fibers significantly enhances the compressive strength of the specimens. The optimal compressive strength is achieved with bamboo fiber lengths of 12 mm. This enhancement can be attributed to the fibers' ability to restrict crack propagation and increase porosity [46,47]. However, if porosity increases beyond the capacity to limit crack growth, the compressive strength will decrease. Conversely, a manageable increase in porosity, while effectively limiting crack extension, can improve compressive strength [48].

Fig 6 illustrates that the capacity of long fibers to enhance the compressive strength of the matrix is significantly greater than that of short fibers. This difference arises primarily from the fact that short fibers contribute to a suboptimal pore structure, which can result in considerable adverse stress. In contrast, the incorporation of long fibers leads to a more effective hoop effect, enhancing the overall integrity of the matrix. However, it is important to note that excessive fiber length may lead to uneven mixing and the formation of agglomerates, which can concentrate local stresses within the matrix. This concentration can influence the damage mode of the specimen and adversely affect overall compression

performance [42,49,50]. Additionally, Fig 6 indicates that the ability of bamboo fibers to enhance the compressive strength of the matrix is not easily discernible at low dosages. Moreover, the compressive strength of the specimen is significantly reduced when the dosage is excessively high and prolonged. This reduction is primarily due to the fact that fiber over-dispersion at lower dosages is more prone to creating unfavorable pressure conditions. Conversely, an excessive quantity of fibers, especially when both numerous and lengthy, can lead to poor dispersion and the formation of agglomerates, ultimately increasing the porosity of the matrix [37,49].

## 3.3. Destruction patterns of BFRPGCs

Fig 7(a) illustrates the flexural strength failure of the specimens. Upon applying the load, the control group specimens exhibited the formation of penetrating cracks at their midpoint. This phenomenon can be attributed to the presence of voids within the phosphogypsum matrix. As the loading increased, the cracks propagated from the bottom upwards until the bond within the phosphogypsum matrix could no longer withstand the additional stress, resulting in specimen failure. The incorporation of bamboo fibers resulted in a random distribution of fibers that provided support to the gypsum matrix. Upon the formation of cracks, the bamboo fibers within the phosphogypsum matrix facilitated a "bridging effect," effectively constraining the propagation of crack growth and enhancing the flexural strength of the specimens. This transformation reflects a shift in damage behavior from brittle to ductile in phosphogypsum cementitious materials (PGCs) to bamboo fiber-reinforced phosphogypsum cementitious materials (BFRPGCs). Upon reaching the maximum maximum load capacity, the samples in the blank group underwent brittle damage and sudden fracture, while the samples with added bamboo fibers underwent continuation damage and initiated the process of cracking [37]. This change is attributed to the bonding network established by the bamboo fibers, which enhances the continuity of the matrix, allowing for more effective regulation of crack width. Furthermore, as depicted in Fig 7(a), the majority of the observed cracks exhibit an oblique curvature, which can be explained by the extension of cracks along the porous zones. When subjected to bending loads, the upper

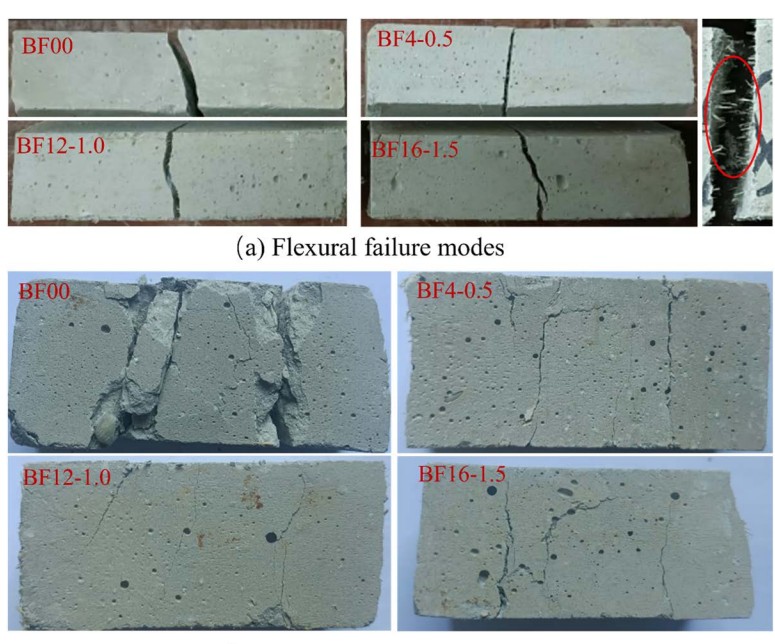

(a) Flexural failure modes

(b) Compressive failure modes

**Fig 7. Damage pattern of the specimen.**

load-bearing region of the specimen experiences compression, while the area between the two lower load-bearing points is subjected to tensile stresses. The integration of bamboo fibers into phosphogypsum-based composite cementitious materials enhances their tensile modulus, resulting in anisotropic behavior and improved tensile capacity in the lower portion of the matrix.

Fig 7(b) illustrates the damage morphology of phosphogypsum composites (PGCs) and bamboo fiber-reinforced phosphogypsum composites (BFRPGCs) under vertical pressure. The results indicate that samples without added bamboo fibers exhibit fractures and partial spalling. When the pores within the phosphogypsum matrix experience longitudinal compression, stress concentration occurs, leading to the formation of transverse cracks that gradually penetrate the material. As the load continues to increase, these cracks extend outward, ultimately compromising the integrity of the specimens once they breach the surface. In contrast, the specimens containing bamboo fibers demonstrate a different fracture behavior, characterized by vertical cracking primarily at the edges of the compression zone. The incorporation of fibers allows the samples to undergo longitudinal compression while being supported transversely by the bridging effect of the bamboo fibers. This reinforcement significantly mitigates the propagation of cracks in the longitudinal direction. Upon reaching the ultimate load, the decrease in load is minimal, and the specimens exhibit a "failure without separation" behavior.

### 3.4. Microanalysis

In this study, the microstructure of bamboo fiber-reinforced phosphogypsum composites (BFRPGCs) was analyzed using scanning electron microscopy (SEM) to investigate the hydration reaction process. As illustrated in Fig 8 (a), the hydration products of BFRPGCs primarily consist of calcium silicate hydrate (C-S-H) gels, calcite alumina (AFt), and a substantial quantity of calcium sulfate dihydrate ($CaSO_4 \cdot 2H_2O$). Furthermore, in the control samples, a multitude of pores is evident in Fig 8 (a), resulting from the extensive spaces between the massive calcium sulfate dihydrate crystals and the expansion of the hydration products (e.g., AFt). In subsequent phases, this phenomenon may lead to the formation of minor fissures [51]. Fig 8 (b) illustrates that in the specimen doped with bamboo fibers, the fibers are embedded within the matrix, with their surfaces extensively adhered to hydration products.

This observation suggests a favorable bond between the matrix and the bamboo fibers, resulting in the formation of a relatively dense interfacial transition zone (ITZ) and an improved pore structure. This robust interaction effectively constrains the relative movement between the bamboo fibers and the matrix when an external load is applied [52,53], while simultaneously impeding the swelling that occurs as a result of the formation and development of hydration products. As illustrated in Fig 7(b), the hydration products are firmly attached to the surfaces of the bamboo fibers. This strong adhesion can be attributed to two factors: the large specific surface area and rough texture of the bamboo fibers [54], as previously discussed, along with the fibers' favorable hydrophilicity [55], which enables them to absorb and retain a portion of the free water, facilitating the precipitation of hydration products on their surfaces.

Fig 9 presents the scanning electron microscopy (SEM) images of bamboo fiber-reinforced phosphogypsum composites (BFRPGCs) with varying degrees of bamboo fiber doping. Hardened phosphogypsum is characterized as a porous material; consequently, the interior of the phosphogypsum composites contains numerous pores when no fibers are incorporated, as illustrated in Fig 9(a). The introduction of bamboo fibers enhances the hydrophilicity of the composite, promoting a more effective integration with the phosphogypsum matrix and resulting in an increased internal density in the resulting BFRPGCs. Fig 9 further illustrates that when the amount of bamboo fibers is insufficient, they fail to disperse thoroughly and uniformly within the gypsum matrix, leading to greater distances between the fibers (Fig 9(b)). In contrast, with a moderate bamboo fiber content, the fibers are evenly distributed throughout the matrix without entanglement or agglomeration, as demonstrated in Fig 9(c). However, at high bamboo fiber concentrations, the dispersion capacity of the fibers within the matrix becomes inadequate, resulting in tendencies to cross, tangle, and agglomerate, yielding an uneven distribution (refer to Fig 9(d)). The hydrophilicity of bamboo fibers facilitates the precipitation and crystallization of

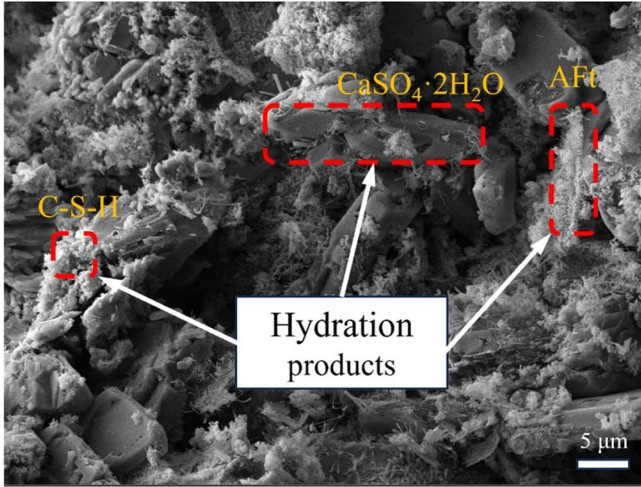

(a) SEM image of PGCs

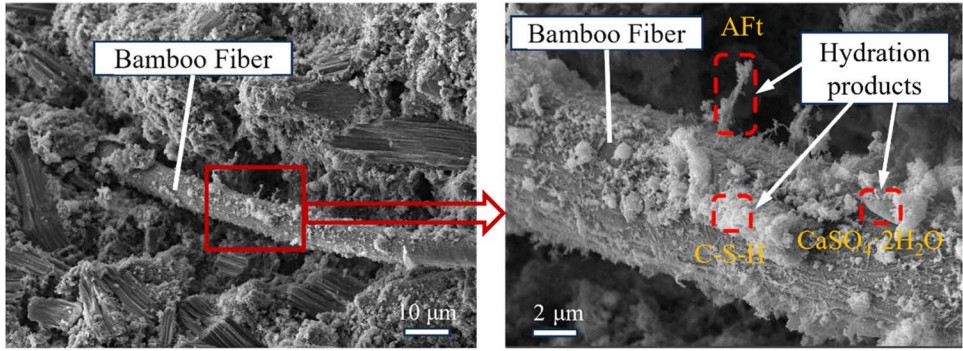

(b) SEM image of BFRPGCs

**Fig 8. Test image of the sample.**

calcium sulfate dihydrate on their surfaces, which enhances adhesion between the fibers and the phosphogypsum matrix. This interaction results in a more compact internal structure of the BFRPGCs, thereby improving the bridging effect of the fibers [56]. Consequently, additional doping with bamboo fibers is required to address the voids and imperfections present within the specimen. Conversely, excessive doping leads to agglomeration, characterized by comparatively minimal phosphogypsum content in these irregularly distributed zones. Moreover, the surfaces of the fibers exhibit a reduced quantity of hydration products, which ultimately diminishes matrix density and compromises the strength of the BFRPGCs.

### 3.5. Mechanism of mechanical properties strengthening of BFEPGCs

The model of bamboo fiber reinforced phosphogypsum composites (BFRPGCs) with varying bamboo fiber lengths at a constant dosage is illustrated in Fig 10. The incorporation of short bamboo fibers facilitates a degree of bridging; however, their limited length leads to rapid displacement following crack formation within the BFRPGCs (see Fig 10 (a)). As a result, the observed enhancement in strength is relatively modest. In contrast, with the increase in bamboo fiber length, the bonding between the fibers and the phosphogypsum matrix becomes more robust, effectively preventing fiber pullout and crack propagation, thereby enhancing strength (Fig 10 (b)). Nevertheless, as the length of the bamboo fibers continues to increase, their dispersion ability within the slurry becomes increasingly constrained, leading to an uneven

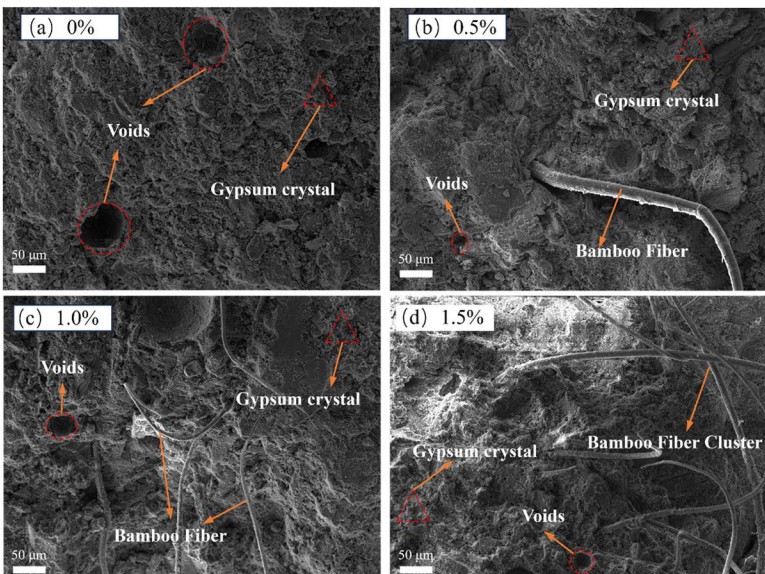

**Fig 9. SEM images of BFRPGCs specimens at different doping levels.**

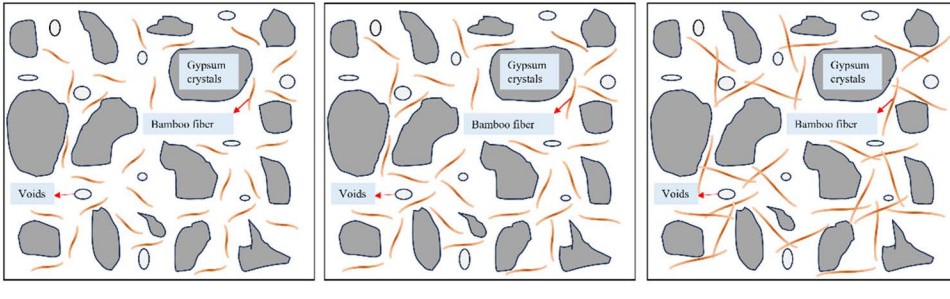

(a) BF with a length of 4mm   (b) BF with a length of 12mm   (c) BF with a length of 16mm

**Fig 10. Model of BFRPGCs with different bamboo fiber lengths at the same doping level.**

distribution characterized by clumps, tangles, and intersections (as illustrated in Fig 10 (c)). Upon hardening of the slurry, this uneven distribution of phosphogypsum in these areas generates stress concentration points, ultimately diminishing the strength of the specimens. Moreover, if the length of the bamboo fibers is insufficient to reach the critical length, the phosphogypsum matrix may be incapable of effectively transferring load to the fibers, resulting in a reduction in the flexural strength of the specimens. When the length of the bamboo fibers is moderate and meets the critical length, they can effectively share the load and provide adequate deformation capacity, thereby significantly enhancing the flexural strength of the specimens [57].

The model of bamboo fiber reinforced polymer composites (BFRPCs) with different bamboo fiber doping of the same length can be found in Fig 11. When the bamboo fiber doping is less, the fibers in the matrix are more dispersed (as in Fig 11(a)), and the bond generated between the fibers and the matrix is not readily apparent, resulting in a relatively modest improvement in strength. As the amount of doping increases, the bamboo fibers become more evenly distributed throughout the matrix (see Fig 11(b)). This results in a stronger bond between the fibers and the matrix, which enhances the

material's overall strength. When the bamboo fiber doping is excessive, the bamboo fibers exhibit inadequate dispersion within the slurry, resulting in cross-links, tangles, and an uneven distribution (Fig 11(c)). It increases the number of pores and defects within the matrix, significantly reducing the material's strength. It can thus be concluded that the addition of moderate amounts of bamboo fibers to the matrix results in the uniform attachment of hydration products to the surface of the fibers. It effectively increases the interaction force between the fibers and the matrix, enhancing the mechanical properties of the BFRPGCs [4].

The phosphogypsum matrix is observed to exhibit brittleness under bending loads in transverse tension (Fig 12(a)). The addition of bamboo fibers to the phosphogypsum matrix serves to prevent the development of cracks through the phenomenon of "bridging effect" (Fig 12(b)). The bamboo fibers within the specimens were selected for analysis, and the resulting loads are illustrated in Fig 14(a). The combined force ($P$) can be divided into two components: the tensile force ($P_1$) along the fiber direction ($P_1 = P \cos \alpha$) and the tensile force ($P_2$) perpendicular to the fiber direction ($P_2 = P \sin \alpha$), representing the adhesion force between the bamboo fibers and the phosphogypsum matrix. When the angle $\alpha$ between the bamboo fibers and the transverse direction $P$ is 0°, $P_1 = P$, and $P_2 = 0$, the bamboo fibers are subjected to a tensile force of $P$. Conversely, when $\alpha$ is 90°, $P_1 = 0$, and $P_2 = P$, the bamboo fibers do not undergo any tensile strain. When the bamboo fibers are distributed uniformly in the matrix, the external force is applied, the stress on the specimen is more uniform, the cracks in the phosphogypsum matrix are effectively limited, and the flexural strength is increased. However, when the bamboo fibers on the specimen are agglomerated, the specimen will exhibit a stress concentration phenomenon after being subjected to an external force. The microcracks in the phosphogypsum matrix will only be partially limited, and the flexural strength will only be increased to a certain extent.

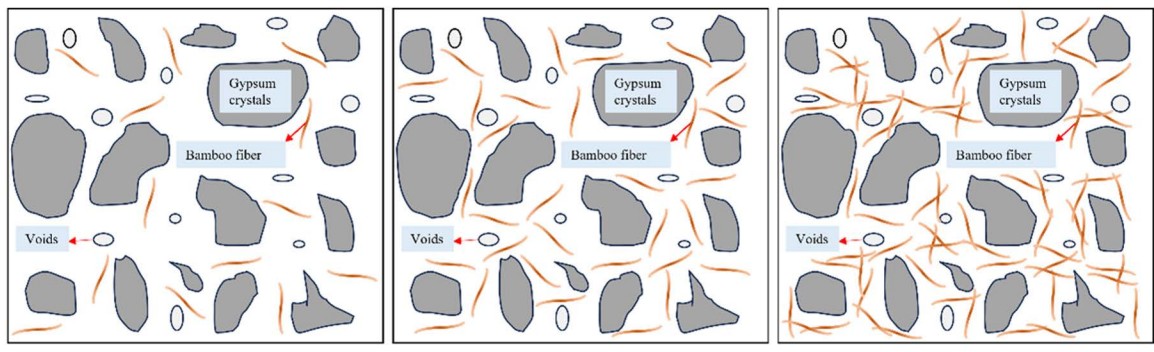

(a) Bamboo fiber with 0.5ωt%    (b) Bamboo fiber with 1.0 ωt %    (c) Bamboo fiber with 1.5 ωt %

**Fig 11. Model of BFRPGCs with different bamboo fiber blends of the same length.**

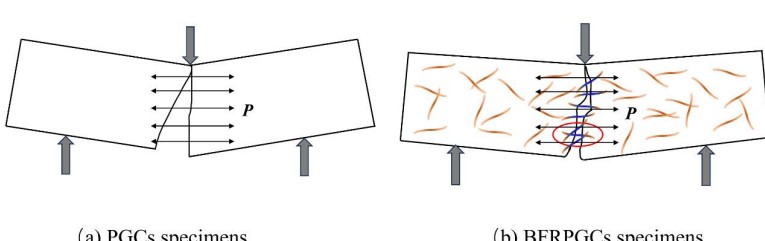

(a) PGCs specimens    (b) BFRPGCs specimens

**Fig 12. Analysis of flexural strength tests.**

When the blank group specimens were subjected to longitudinal compression, a multitude of longitudinal cracks emerged on the surface, resulting in the two sides being crushed and the surface assuming a "saddle-shaped" appearance (Fig 13(a)). The addition of bamboo fibers resulted in the suppression of longitudinal cracks due to the bridging effect of the fibers (Fig 13(b)). A single bamboo fiber from the specimen was selected for examination, and the loading is illustrated in Fig 14(b) below. The longitudinal pressure ($F$) can be divided into two components: the pressure ($F_1$) along the direction of the bamboo fibers ($F_1 = F\cos\beta$) and the pressure ($F_2$) perpendicular to the direction of the bamboo fibers ($F_2 = F\sin\beta$). When the angle $\beta$ between the bamboo fiber and the pressure $F$ is 0°, $F_1 = F$ and $F_2 = 0$, the direction of the bamboo fiber is parallel to the direction of the force, and the pressure on the bamboo fiber is the $F$ pressure. When $\beta$ equals 90°, $F_1$ is equal to 0, and $F_2$ is equal to $F$. In this case, the direction of the bamboo fiber is perpendicular to the direction of the force, which is primarily resisted by the shear properties of the bamboo fiber.

When subjected to longitudinal compression, specimens exhibiting a uniform distribution of bamboo fibers demonstrated a marked increase in resistance to crack development, thereby enhancing their compressive strength. In contrast, the presence of non-uniform distributions of bamboo fibers leads to the formation of stress concentration phenomena, which ultimately diminishes compressive strength.

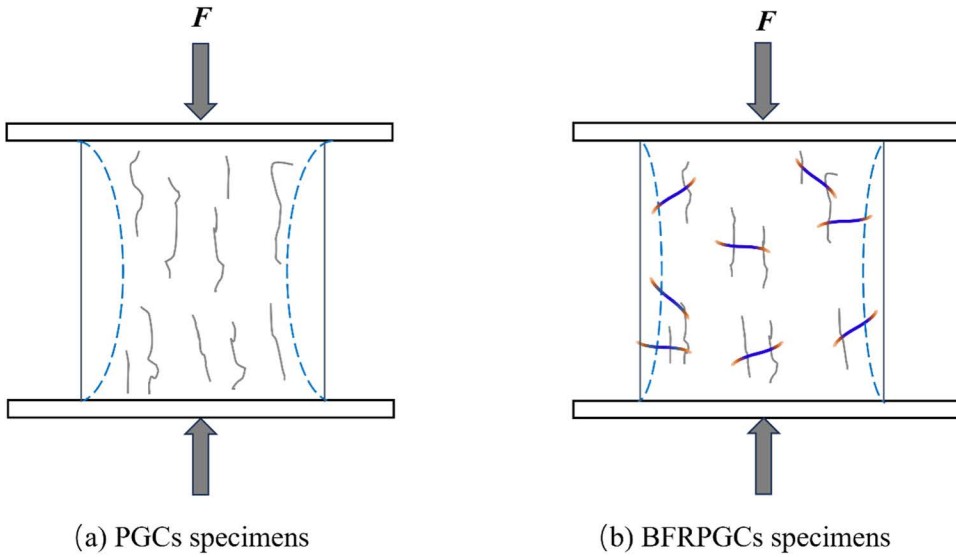

(a) PGCs specimens    (b) BFRPGCs specimens

**Fig 13. Analysis of compressive strength tests.**

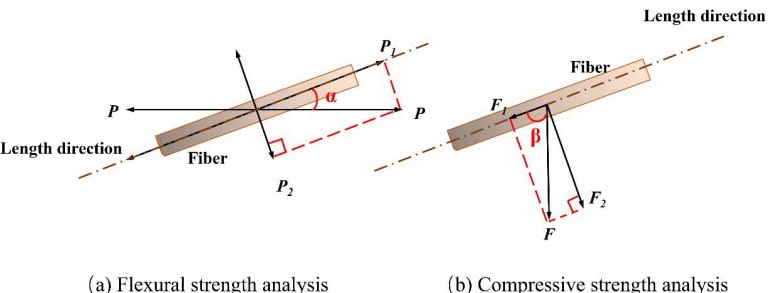

(a) Flexural strength analysis    (b) Compressive strength analysis

**Fig 14. Strength test analysis of BFRPGCs.**

## 4. Conclusions

This study investigates the mechanical properties of raw bamboo fiber-reinforced phosphogypsum-based composite cementitious materials (BFRPGCs) and elucidates their reinforcement mechanisms. The following conclusions can be drawn from the investigation:

(1) An increase in the water-cement ratio results in a reduction of the mechanical properties of BFRPGCs. This decline can be attributed to several interrelated factors, including increased porosity, incomplete hydration, the presence of air bubbles, and delamination.

(2) The incorporation of bamboo fibers significantly enhances the mechanical properties of BFRPGCs. Among the variables tested, the compressive and flexural strengths of BFRPGCs at 28 days were maximized with a fiber length of 12 mm and a doping amount of 1.0%. These strengths exhibited increases of 123.73% and 169.82%, respectively, when compared to the control group.

(3) The SEM results indicate that raw bamboo fibers demonstrate strong adhesion to the phosphogypsum composite matrix. This adhesion is primarily attributed to the rough surface texture and high water absorption capacity of natural bamboo fibers, which facilitate the attachment of a substantial quantity of hydration products to the fiber surfaces.

(4) The mechanism underlying the enhancement of mechanical properties in BFRPGCs was investigated through an analysis of the stress damage patterns associated with bamboo fibers. It was determined that fibers of moderate length provide a more uniform distribution within the composite, effectively inhibiting crack propagation in the phosphogypsum composites and exhibiting a more pronounced "bridging effect."

## 5. Challenges and prospects

Although this study demonstrated the positive effects of raw bamboo fibers on the properties of phosphogypsum-based cementitious materials, a critical challenge remains in translating these results from laboratory conditions to practical industrial applications, especially with regard to ensuring uniform fiber dispersion. Experimentally, we have improved fiber dispersion by optimizing the fiber pretreatment processes (e.g., drying, artificial dispersion, etc.). However, under industrial-scale production conditions, achieving uniform fiber distribution remains a significant challenge. Therefore, future research should focus on the use of efficient mixing technologies, such as high-shear mixers or ultrasonic dispersing equipment, to further improve fiber dispersion and distribution uniformity. Additionally, given the natural characteristics of raw bamboo fibers, their hydrophilicity and hygroscopicity may impact the large-scale production process. As such, further process optimization is essential for future work. We believe that with continuous improvements in both the production process and equipment, the application of raw bamboo fibers in phosphogypsum-based cementitious materials holds great promise and can contribute to the development of the green building materials industry.

## Acknowledgments

We would like to thank all the reviewers and editors for their valuable comments, which greatly improved this manuscript.

## Author contributions

**Conceptualization:** Xian Fu.

**Formal analysis:** Xian Fu.

**Funding acquisition:** Dewen Kong, Yuan Li.

**Investigation:** Yongfa Wang.

**Methodology:** Dewen Kong.

**Project administration:** Peng Liu.

**Resources:** Peng Liu, Dewen Kong.

**Supervision:** Xian Fu.

**Writing – original draft:** Xian Fu.

**Writing – review & editing:** Peng Liu.

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
