## [Decision Letter · Decision Letter 0]

15 Dec 2024

PONE-D-24-56170Mechanical properties of raw bamboo fiber-reinforced phosphogypsum-based composite cementitious materials and their strengthening mechanismPLOS ONE

Dear Dr. Peng Liu,

Thank you for submitting your manuscript to PLOS ONE. After careful consideration, we feel that it has merit but does not fully meet PLOS ONE’s publication criteria as it currently stands. Therefore, we invite you to submit a revised version of the manuscript that addresses the points raised during the review process.

Please submit your revised manuscript by Jan 29 2025 11:59PM.. If you will need more time than this to complete your revisions, please reply to this message or contact the journal office at plosone@plos.org . Please include the following items when submitting your revised manuscript:

We look forward to receiving your revised manuscript.

Kind regards,

Solomon Oyebisi, PhD

Academic Editor

PLOS ONE

Journal Requirements:

“This study was supported by the Science and Technology Plan Project Guizhou Provincial, China (Qiankehejichu-ZK[2023] General 067); the Innovation Fund of Guizhou University Survey and Design Institute Co., Ltd. (Guidakancha [2022]05), and the Open Laboratory Project of Guizhou University (SYSKF2024-006).”

Reviewers' comments:

Reviewer's Responses to Questions

**Comments to the Author**

1. Is the manuscript technically sound, and do the data support the conclusions?

Reviewer #1: Yes

Reviewer #2: Yes

2. Has the statistical analysis been performed appropriately and rigorously? 

Reviewer #1: Yes

Reviewer #2: Yes

3. Have the authors made all data underlying the findings in their manuscript fully available?

Reviewer #1: Yes

Reviewer #2: Yes

4. Is the manuscript presented in an intelligible fashion and written in standard English?

Reviewer #1: Yes

Reviewer #2: Yes

5. Review Comments to the Author

Reviewer #1: 1- While the paper highlights the potential of bamboo fibers for reinforcing phosphogypsum, the novelty should be better articulated by comparing results with existing materials like polypropylene or steel fibers.

2- The selection of the optimal bamboo fiber length (12 mm) and doping amount (1.0%) lacks a clear explanation of whether other combinations might provide comparable or better results.

3- The one-way experimental design is appropriate but limits the exploration of interactions between parameters. A factorial design could provide deeper insights.

4- The study does not mention the use of statistical tools to validate the significance of observed trends in mechanical properties. Adding ANOVA or regression analysis could strengthen the reliability of findings.

5- The mechanical properties of control specimens are referenced but not thoroughly analyzed to demonstrate the incremental benefits of bamboo fibers.

6- The SEM analysis is informative but would benefit from quantitative metrics, such as porosity measurements or fiber-matrix adhesion strength, to substantiate qualitative observations.

7- The long-term performance and durability of BFRPGCs under environmental conditions (e.g., moisture, freeze-thaw cycles) are not addressed, which is critical for construction materials.

8- The paper does not discuss potential challenges in scaling the laboratory findings to industrial applications, particularly with uniform fiber dispersion.

9- The effect of excessive bamboo fiber concentration on mechanical properties due to agglomeration is noted but not quantified. Including dispersion indices would add rigor.

10- The mechanical performance of BFRPGCs could be contextualized better by comparing it directly with other fiber-reinforced cementitious materials.

11- The environmental benefits of using bamboo fibers and phosphogypsum are implied but not quantified. Including a life-cycle analysis or CO₂ emission savings could be valuable.

12- While hydration products are identified, their specific role in enhancing fiber-matrix interaction should be elaborated with additional microstructural data.

13- The distinction between brittle and ductile failure modes is described but not supported with stress-strain curves or energy absorption metrics.

14- The choice of water-cement ratio range (0.250–0.450) seems arbitrary. Justifying this range based on literature or preliminary studies would strengthen its credibility.

15- Given the scope and content of this paper, it may benefit from considering the following related works:

https://doi.org/10.1061/(ASCE)MT.1943-5533.0004528

https://doi.org/10.1016/j.conbuildmat.2021.125874

https://doi.org/10.1016/j.conbuildmat.2021.124645

Reviewer #2: The presented study is an original study on the usability of bamboo fiber and phosphogypsum as additives in cementitious materials production. Within the scope of the study, the required properties of cementitious materials were examined in detail and the results were presented understandably. The structure of the study, the methods used, and the evaluation of the results are problem-free. My only negative opinion about the study is that the references are not current enough. If it is possible, please supplement the manuscripts below.

*https://doi.org/10.1080/15440478.2022.2162186

*https://doi.org/10.3390/buildings12091450

*https://doi.org/10.1016/j.heliyon.2024.e24313

*https://doi.org/10.1016/j.heliyon.2024.e24313

*https://doi.org/10.1016/j.matpr.2023.03.010

Then, the journal could put your manuscript into its current archive.

Best wishes.

6. PLOS authors have the option to publish the peer review history of their article (what does this mean? ). If published, this will include your full peer review and any attached files.

**Do you want your identity to be public for this peer review?** For information about this choice, including consent withdrawal, please see our Privacy Policy .

Reviewer #1: No

Reviewer #2: No

---

## [Author Response · Author response to Decision Letter 1]

3 Jan 2025

Responses to reviewers' comments

Dear Editors and Reviewers,

Thank you for your letter and for the reviewers' comments concerning our revised manuscript entitled "Mechanical properties of raw bamboo fiber-reinforced phosphogypsum-based composite cementitious materials and their strengthening mechanism" (PONE-D-24-56170). Those comments are all valuable and very helpful for revising and improving our paper, as well as the important guiding significance to our researches. We have studied comments carefully and have made correction which we hope to meet with approval. Revised portion is marked in red in the paper. The main corrections in the paper and the response to their viewer's comments are as flowing:

Reviewer #1:

1. Comment: While the paper highlights the potential of bamboo fibers for reinforcing phosphogypsum, the novelty should be better articulated by comparing results with existing materials like polypropylene or steel fibers.

Response: Thank you for your careful review and valuable comments on the paper. Firstly, as noted in the introduction, I acknowledge that the list of adverse effects associated with various contemporary fibres, such as polypropylene and steel fibres, may not be exhaustive. To address this, a further comparison will be made between the performance of bamboo fibre and polypropylene fibre and steel fibre in terms of different performance indexes, as illustrated in a table:

Table 1

Performance of bamboo fibre with polypropylene fibre, steel fibre and polyvinyl alcohol fibre on different performance indicators

Performance indicators Bamboo fibres Polypropylene fibres Steel fibres Polyvinyl alcohol fibres

Mechanical performance High tensile strength, good toughness, enhanced material crack resistance Lower tensile strength for improved material resistance to cracking Superior tensile strength to increase the overall strength of the material Good tensile properties, but weak compared to steel fibres

Environmental benefit Renewable Energy, biodegradable, low carbon footprint Petroleum-based material, non-degradable, negative impact on the environment High energy consumption, difficult recycling, higher carbon emissions Environmentally friendly materials, biodegradable, low carbon footprint

Cost effectiveness Low-cost, resourceful and particularly suited to large-scale production Medium cost, widely used but costly in the long term High cost, higher production and transport costs Higher production costs, especially in large-scale utilisation

References Zhang et al. (2020), Liu et al. (2021) Silva et al. (2018), Mahmood et al. (2020) Li et al. (2017), Mohamed et al. (2019) Zhang et al. (2022), Chen et al. (2020)

Secondly, within the ambit of the 'bamboo instead of plastic' research, the utilisation of bamboo fibre-reinforced phosphogypsum, as an emerging building material, has the potential to amalgamate the environmental properties of phosphogypsum with the elevated strength of bamboo fibres. In comparison to conventional polypropylene and steel fibre reinforcements, bamboo fibre-reinforced phosphogypsum exhibits distinctive advantages in terms of environmental sustainability, resource efficiency, and compatibility with biodegradable materials. In response to your suggestion, we will further clarify and mark in red the introductory paragraph, which will be reworked as follows:

To enhance the mechanical properties of phosphogypsum and broaden its construction applications, numerous studies have been conducted globally. However, practical engineering applications face challenges: steel fibers are prone to corrosion when exposed to air [18, 19], polypropylene (PP) fibers exhibit low tensile strength [20, 21], polyvinyl alcohol fibers are costly [22, 23], and glass and asbestos fibers pose health hazards [24, 25]. In contrast, plant fibers are natural polymer materials with specific chemical compositions [26, 27], drawing significant attention due to their safety, availability, biodegradability, and potential for sustainable regeneration [28-30]. Among these, bamboo fiber—a rapidly renewable plant fiber—exhibits mechanical properties comparable to traditional fiber-reinforced cement, significantly enhancing the toughness of composite cementitious materials [31, 32]. Moreover, bamboo fibre-reinforced phosphogypsum has been demonstrated to exhibit distinct advantages over conventional fibre-reinforced materials with regard to environmental friendliness, resource and sustainability, and compatibility with biodegradable materials.

2. Comment: The selection of the optimal bamboo fiber length (12 mm) and doping amount (1.0%) lacks a clear explanation of whether other combinations might provide comparable or better results.

Response: Thank you for your careful review. Regarding your question “The selection of the optimal bamboo fiber length (12 mm) and doping amount (1.0%) lacks a clear explanation of whether other combinations might provide comparable or better results.” we have revisited the experimental design and provided an explanation.

In the experiment, the variation in length of bamboo fibre was set at 4mm, 8mm, 12mm, and 16mm, with the fibre amount for each length set at four levels (0, 0.50, 1.0, 1.5wt%), as evidenced by the experimental results:

1� A bamboo fibre length of 12 mm has been shown to optimise the properties of the composite while maintaining adequate mechanical strength and stability. The addition of longer bamboo fibres (16 mm) has been found to enhance the strength to a certain extent, however, it has also been observed to affect the dispersion of the fibres and the homogeneity of the material. In contrast, shorter fibres (4 mm) have been found to be ineffective in providing the desired reinforcing effect of bamboo fibres. Consequently, the 12 mm length is regarded as a balanced choice.

2� At a doping level of 1.0 wt%, the bamboo fibres were found to enhance the mechanical properties of the composites to a significant extent, without exerting an excessive negative impact on the processing properties of the materials. Conversely, higher (1.5 wt%) or lower (0.50 wt%) doping levels resulted in a decline in mechanical properties.

3� Following the experimental study, it was determined that the optimal mechanical properties of BFRGCs were attained with a bamboo fibre length of 12 mm and a doping level of 1.0 wt%, as illustrated below:

Fig. 5. Variation of flexural strength of BFRPGCs.

Fig. 6. Variation of compressive strength of BFRPGCs.

3. Comment: The one-way experimental design is appropriate but limits the exploration of interactions between parameters. A factorial design could provide deeper insights.

Response: Thank you for your careful review and valuable comments on our work. We appreciate your suggestions regarding the experimental design. We agree with your observation that “the one-way experimental design is appropriate but limits the exploration of interactions between parameters, and a factorial design could provide deeper insights.” Due to time and resource constraints in the current study, we chose a one-factor experimental design to focus on the effects of bamboo fiber length and dosage on the mechanical properties of phosphogypsum-based cementitious materials. This limited our ability to perform ANOVA or explore interactions between parameters. We understand that a factorial design would offer a more comprehensive understanding, particularly in terms of multi-parameter interactions. We apologize for not being able to implement a factorial design in this study. However, in future research, we plan to expand the experimental design to incorporate a factorial approach, which will allow us to better explore the interactions among factors and conduct the relevant statistical analyses. This will provide deeper insights into the combined effects of individual parameters on the experimental results.

4. Comment: The study does not mention the use of statistical tools to validate the significance of observed trends in mechanical properties. Adding ANOVA or regression analysis could strengthen the reliability of findings.

Response: Thank you for your careful review of our study. We greatly appreciate your questions regarding the statistical analysis tools (e.g., ANOVA and regression analysis). We fully recognize the value of these methods in verifying the significance of trends in mechanical properties and in enhancing the reliability of the study. In the current study, statistical methods such as analysis of variance (ANOVA) or regression analysis were not employed because our experimental design was based on a one-way approach, focusing primarily on the effect of each individual variable (such as the length of raw bamboo fibers or the amount of blending) on the mechanical properties. Although we performed multiple repetitions in our experiments and observed relatively consistent trends across the experimental groups, which led us to believe that the results were reliable, we did not use statistical tools to quantify the significance of these trends.

We apologize for this omission and fully acknowledge the importance of your suggestions in further enhancing the rigor and credibility of our study. For future research, we plan to adopt more comprehensive experimental designs, including multifactorial and factorial approaches, which will enable us to apply statistical tools (such as ANOVA and regression analysis) to validate the significance of the results and strengthen the overall credibility of our findings.

Once again, we are grateful for your valuable comments, which have provided crucial direction for the improvement of our future research. We will incorporate your suggestions into our experimental design and ensure that appropriate statistical analyses are conducted in future studies. We look forward to receiving your guidance again in the future.

5. Comment: The mechanical properties of control specimens are referenced but not thoroughly analyzed to demonstrate the incremental benefits of bamboo fibers.

Response: Thank you for your valuable suggestion. We appreciate your identification of the issues related to the analysis of the mechanical property control samples. We agree that this section requires further discussion to better highlight the positive effect of bamboo fibers. In the current study, while we did mention the mechanical properties of the control (blank) samples, the primary focus was on evaluating the impact of bamboo fibers on the mechanical properties of phosphogypsum-based cementitious materials. As a result, our description of the control samples was relatively brief. We now recognize that a more detailed analysis of the matrix strength without the addition of bamboo fibers is crucial to more clearly demonstrate the benefits provided by the bamboo fibers. We apologize for this omission and will provide a more comprehensive analysis of the control samples in the revised version.

Specifically, we will include a detailed data analysis of the control samples, focusing on mechanical property indices, and compare these with the samples containing bamboo fibers to better illustrate the gain effect of the bamboo fibers. This will not only make the results more thorough and persuasive, but will also clarify the actual contribution of bamboo fibers to the material properties.

Once again, we greatly appreciate your insightful comments. We have added the necessary analyses to the manuscript, and these additions are highlighted in red. The revised sections are provided below:

1� Fig. 5 illustrates the variation in flexural strength of bamboo fiber-reinforced phosphogypsum composites (BFRPGCs) as a function of bamboo fiber length and doping amount. Without the addition of bamboo fibers, the 7-day and 28-day flexural strengths of PGCs were 4.31 MPa and 4.95 MPa, respectively. After the incorporation of bamboo fibers, the flexural strength of BFRPGCs gradually increased with the increase in fiber content, while the length of the bamboo fibers remained constant.

2� Fig. 6 illustrates the relationship between the compressive strength of Bamboo Fiber Reinforced Geopolymer Composites (BFRPGCs), the length of bamboo fibers, and the amount of doping. Without the addition of bamboo fibers, the 7-day and 28-day compressive strengths of PGCs were 16.20 MPa and 23.43 MPa, respectively. After the addition of bamboo fibers, the 7-day compressive strength of BFRPGCs increased gradually with the increase in fiber content, provided that the length of the bamboo fibers remained unchanged and did not exceed 12 mm.

6. Comment: The SEM analysis is informative but would benefit from quantitative metrics, such as porosity measurements or fiber-matrix adhesion strength, to substantiate qualitative observations.

Response: Thank you for your valuable comment. We greatly appreciate your suggestion to incorporate quantitative metrics into the SEM analysis. While SEM images provide important qualitative information, we agree that including quantitative analyses (e.g., porosity measurements or quantification of fiber-matrix bond strength) would enhance the credibility of our findings and better support our observations. In this study, the SEM analysis focused on examining the microscopic morphology of the samples and describing the fiber-matrix interface. However, due to the limitations of the experimental design, we did not conduct quantitative measurements of porosity or fiber-matrix bond strength. We apologize for this limitation and recognize that your suggestions are essential for improving the rigor of our study.

As you pointed out, porosity is an important factor influencing the mechanical and adhesive properties of the phosphogypsum matrix. In future studies, we plan to extend our analysis by incorporating porosity measurements, tests of fiber-matrix bond strength, and other relevant quantitative metrics. These additional analyses will allow us to validate the trends observed in the SEM images more comprehensively and further strengthen the reliability of our conclusions.

Thank you again for your professional guidance. We have revisited the paper and made improvements to the SEM images. The revised version, with changes marked in red, is shown in the figure below:

Fig. 9. SEM images of BFRPGCs specimens at different doping levels.

7. Comment: The long-term performance and durability of BFRPGCs under environmental conditions (e.g., moisture, freeze-thaw cycles) are not addressed, which is critical for construction materials.

Response: Thank you for your valuable comments. We take your question regarding the long-term performance and durability of bamboo fiber-reinforced phosphogypsum-based cementitious materials (BFRPGCs) under environmental conditions (e.g., humidity, freeze-thaw cycles, etc.) very seriously. We fully agree that these factors are critical for the practical application of building materials.

In the current study, we only conducted short-term performance tests, primarily focusing on a 28-day curing period. The main objective of these experiments was to evaluate the initial impact of bamboo fibers on the mechanical properties of phosphogypsum-based cementitious materials and to provide a preliminary assessment of the material's performance. Due to the limitations of the experimental design, we did not address long-term durability aspects (e.g., freeze-thaw cycles, humidity effects, etc.) or environmental adaptability in depth. We apologize for this limitation and fully understand your concerns regarding the long-term performance and durability of the material. These factors are indeed crucial for the practical use of building materials, especially in varying environmental conditions. As such, we plan to extend our future research to include long-term durability testing, specifically focusing on the performance of BFRPGCs under the influence of humidity, freeze-thaw cycles, and other environmental factors. These follow-up experiments will allow us to more comprehensively assess the long-term stability and durability of bamboo fiber-reinforced phosphogypsum-based materials and provide stronger support for their practical application in engineering projects. In addition, we intend to adopt standardized durability testing methods (e.g., freeze-thaw cycles, hot and humid enviro

---

## [Decision Letter · Decision Letter 1]

1 Apr 2025

Mechanical properties of raw bamboo fiber-reinforced phosphogypsum-based composite cementitious materials and their strengthening mechanism

PONE-D-24-56170R1

Dear Dr. Liu,

We’re pleased to inform you that your manuscript has been judged scientifically suitable for publication and will be formally accepted for publication once it meets all outstanding technical requirements.

Kind regards,

Solomon Oyebisi, PhD

Academic Editor

PLOS ONE

Additional Editor Comments (optional):

Reviewers' comments:

Reviewer's Responses to Questions

**Comments to the Author**

1. If the authors have adequately addressed your comments raised in a previous round of review and you feel that this manuscript is now acceptable for publication, you may indicate that here to bypass the “Comments to the Author” section, enter your conflict of interest statement in the “Confidential to Editor” section, and submit your "Accept" recommendation.

Reviewer #1: (No Response)

Reviewer #2: All comments have been addressed

2. Is the manuscript technically sound, and do the data support the conclusions?

Reviewer #1: (No Response)

Reviewer #2: Yes

3. Has the statistical analysis been performed appropriately and rigorously? 

Reviewer #1: (No Response)

Reviewer #2: Yes

4. Have the authors made all data underlying the findings in their manuscript fully available?

Reviewer #1: (No Response)

Reviewer #2: Yes

5. Is the manuscript presented in an intelligible fashion and written in standard English?

Reviewer #1: (No Response)

Reviewer #2: Yes

6. Review Comments to the Author

Reviewer #1: Upon reviewing the authors' responses and the revised manuscript, the paper has been significantly improved and now meets the publication criteria of the journal. Acceptance of the manuscript is recommended.

Reviewer #2: Since the authors have revised their papers as comments I have given previously, the journal could put their paper into its current archive.

7. PLOS authors have the option to publish the peer review history of their article (what does this mean? ). If published, this will include your full peer review and any attached files.

**Do you want your identity to be public for this peer review?** For information about this choice, including consent withdrawal, please see our Privacy Policy .

Reviewer #1: No

Reviewer #2: No

---

## [Editor Report · Acceptance letter]

PONE-D-24-56170R1

PLOS ONE

Dear Dr. Liu,

I'm pleased to inform you that your manuscript has been deemed suitable for publication in PLOS ONE. Congratulations! Your manuscript is now being handed over to our production team.

Kind regards,

on behalf of

Dr. Solomon Oyebisi

Academic Editor

PLOS ONE